# Comparative Application of BioID and TurboID for Protein-Proximity Biotinylation

**DOI:** 10.3390/cells9051070

**Published:** 2020-04-25

**Authors:** Danielle G. May, Kelsey L. Scott, Alexandre R. Campos, Kyle J. Roux

**Affiliations:** 1Enabling Technologies Group, Sanford Research, Sioux Falls, SD 57104, USA; danielle.may@sanfordhealth.org (D.G.M.); kelsey.scott@sanfordhealth.org (K.L.S.); 2Proteomics Facility, Sanford Burnham Prebys Medical Discovery Institute, La Jolla, CA 92037, USA; arosacampos@sbpdiscovery.org; 3Department of Pediatrics, Sanford School of Medicine, University of South Dakota, Sioux Falls, SD 57105, USA

**Keywords:** proximity-labeling, BioID, TurboID, biotinylation, nuclear pore complex, lamin

## Abstract

BioID is a well-established method for identifying protein–protein interactions and has been utilized within live cells and several animal models. However, the conventional labeling period requires 15–18 h for robust biotinylation which may not be ideal for some applications. Recently, two new ligases termed TurboID and miniTurbo were developed using directed evolution of the BioID ligase and were able to produce robust biotinylation following a 10 min incubation with excess biotin. However, there is reported concern about inducibility of biotinylation, cellular toxicity, and ligase stability. To further investigate the practical applications of TurboID and ascertain strengths and weaknesses compared to BioID, we developed several stable cell lines expressing BioID and TurboID fusion proteins and analyzed them via immunoblot, immunofluorescence, and biotin-affinity purification-based proteomics. For TurboID we observed signs of protein instability, persistent biotinylation in the absence of exogenous biotin, and an increase in the practical labeling radius. However, TurboID enabled robust biotinylation in the endoplasmic reticulum lumen compared to BioID. Induction of biotinylation could be achieved by combining doxycycline-inducible expression with growth in biotin depleted culture media. These studies should help inform investigators utilizing BioID-based methods as to the appropriate ligase and experimental protocol for their particular needs.

## 1. Introduction

Proximity-dependent labeling methods for identifying protein–protein interactions (PPIs) have become an established class of tools developed and utilized by hundreds of researchers within the last decade. These methods include antibody-based approaches such as SPLAAT and EMARS [1,2], the ascorbate peroxidase-based approach called APEX [3,4], the biotin-ligase approach BioID [5,6], and, more recently, a pupylation-based approach termed PUP-IT for identifying membrane protein interactions [7]. While these systems have many similarities, their respective limitations restrict their practical applications. For example, the accuracy of the SPLAAT and EMARS methods will heavily depend on the quality of the antibody used and is traditionally restricted to plasma membrane interactions. Recent modifications to these antibody-based approaches has expanded the use of primary antibodies in conjunction with HRP-conjugated secondary antibodies for proximity labeling within the nucleus of fixed cells [8]. The APEX system utilizes an extremely rapid peroxidase-based labeling induced by exposure of cells or tissues to mM levels of H_2_O_2_ in cells preincubated with biotin-phenol and typically requires the use of SILAC to interpret the data due to high background labeling. BioID requires a long labeling period of 15–18 h and does not function well at temperatures below 37 °C, and PUP-IT may not be suitable for interactions within organelles since Pup, a prokaryotic 64-amino acid protein, cannot diffuse across membranes.

We developed and applied the original biotin ligase used for proximity-dependent labeling using a humanized version of the BirA protein from *E. coli* with a R118G mutation [6] that was known to enable promiscuous biotinylation [9,10]. Wild-type BirA selectively biotinylates acetyl coA carboxylase by releasing a primed bioAMP molecule for covalent attachment to a specific lysine [11,12]. The R118G mutation decreases the affinity of BirA* (hereafter referred to as BioID) for both biotin and bioAMP, about 40- and 440-fold compared to the wild-type, respectively [11]. The reduced affinity to bioAMP leads to a dramatically enhanced release of reactive bioAMP molecules from the ligase which covalently labels available lysines on proteins which we demonstrated to occur within a ~10 nm radius [2,6,9,10,13]. Therefore, BioID can be used to map protein networks within live cells. The reduced affinity of the BioID ligase to biotin likely prevents substantive biotinylation without the addition of excess levels of biotin (5–50 μM), thus enabling the ability to induce the onset of biotinylation and thus temporally control the promiscuous labeling to enable selective or comparative studies. We developed a second promiscuous biotin ligase (BioID2) from *A. aeolicus* as a smaller, somewhat more biotin-sensitive alternative to the original BioID ligase [5]. Overall, the BioID method has been cited and/or applied in over 300 articles investigating a wide range of proteins and subcellular domains (for review, see [14,15]).

While BioID/BioID2 have been applied successfully in many model systems including in live cultured cells, yeast [16,17], parasites [18,19,20,21,22,23,24], plants [25,26], and mice [27,28,29,30], most experiments have been performed utilizing a 12–24 h labeling period, with few exceptions labeling for 1 h or 3 h [31,32]. Experiments requiring shorter labeling periods require a faster version of BioID and one that would work well at temperatures well below 37 °C. Two groups have reported versions of BioID that address some or all of these limitations. A *B. subtilis*-based version of BioID [33], called BASU, that was used to map RNA–protein interactions was originally claimed to be 1000 times faster than the original BioID, but these results have not been reproducible [34]. More recently, a directed-evolution variant of BioID, called TurboID, was reported to be substantially faster than the original BioID ligase and capable of functioning effectively at lower temperatures [34]. In that same study, a smaller version of TurboID (termed miniTurbo) was also developed by removing the N-terminus of TurboID. Here, we comparatively evaluate the practical application and labeling radius of TurboID and the original BioID. We use previously validated baits in the nuclear envelope—including Lamin A (LaA), Nup43, and Nup53—in order to reveal optimal practical applications and to better characterize the growing toolbox of proximity-dependent labeling biotin ligases [6,13]. 

## 2. Materials and Methods

### 2.1. Plasmids

3xHA-TurboID and 3xHA-miniTurbo were amplified via PCR from Addgene constructs #107171 and #107172, respectively. Amplified PCR products were inserted into pBabe via In-Fusion Recombination into mycBioID pBabe (Addgene #80901), replacing mycBioID, using EcoRI and XhoI restriction enzyme (RE) sites. LaA, Nup43, Nup53, and Sun2 were amplified by PCR from human cDNA. The PCR products were inserted into the newly made N-terminal 3xHA-TurboID using XhoI and PmeI or C-terminal TurboID-3xHA pBabe puro constructs using NaeI and EcoRI RE sites. 3xHA-BioID was made by amplifying 3xHA from TurboID-3xHA and BioID from mycBioID pBabe puro and two-step In-Fusion cloning. Amplified PCR product was inserted into pBabe puro using EcoRI and XhoI. C-terminal LaA or Nup53 were subsequently inserted using XhoI and PmeI restriction sites. BioID-3xHA was made by cutting out TurboID from –TurboID-3xHA pBabe with EcoRI and BamHI and inserting BioID amplified PCR product. N-terminal fusions of Nup43 or Sun2 were inserted via NaeI and EcoRI RE sites. Intermediate TurboID generations were synthesized at Gene Universal (Newark, DE, USA), PCR amplified, and inserted into 3xHA-BioID pBabe puro by removing BioID with AgeI and XhoI. Dox-inducible 3xHA-TurboID pRetroX pTight construct was made by amplifying 3xHA-TurboID and inserting it into pRetroX pTight via In-Fusion cloning into EcoRI and BamHI sites.

### 2.2. Cell Culture

A549 cells were obtained from the American Type Culture Collection (ATCC; CCL-185™). Stable cell lines for all BioID, TurboID, and miniTurbo constructs were generated using retroviral transduction. HEK293 Phoenix cells (National Gene Vector Biorepository, Indianapolis, IN, USA) were transfected with each construct using Lipofectamine 3000 (Thermo Fisher Scientific, Waltham, MA, USA) per the manufacturer’s recommendation. The transfected cells were incubated at 37 °C for 6 h. After 6 h incubation, the transfected cells were replenished with fresh medium and further incubated at 32 °C for 72 h. The culture media was filtered through a 0.45-μm filter and added to A549 cells along with Polybrene (4 μg/mL; Santa Cruz Biotechnology, Dallas, TX, USA). At 72 h after transduction, puromycin (0.5 μg/mL; Thermo Fisher Scientific) was added to the target cells. The expression of fusion proteins and functional biotinylation was further verified using IF and WB. For cell lines requiring doxycycline (dox)-induction, dox (1 mg/mL) was added either 24 h before supplementation with 50 µM biotin (0 min, 10 min, 1 h, 4 h) or concurrently with 50 µmM biotin (18 h). Dialyzed serum was obtained from Fisher Scientific (Pittsburgh, PA, USA; Gibco, A3382001). The stable cell lines were maintained in 5.0% CO_2_ at 37 °C in DMEM (HyClone, Logan, UT, USA) supplemented with 10% fetal bovine serum (FBS). All cells were tested monthly for mycoplasma contamination.

### 2.3. Immunofluorescence

Cells grown on glass coverslips were fixed in 3% (wt/vol) paraformaldehyde/phosphate-buffered saline (PBS) for 10 min and permeabilized by 0.4% (wt/vol) Triton X-100/PBS for 15 min. For labeling fusion protein purposes, a mouse anti-hemagglutinin primary antibody was used (HA; 1:1000; 12CA5; Covance, Princeton, NJ, USA). The primary antibody was detected using Alexa Fluor 488–conjugated goat anti-mouse (1:1000; A11029; Thermo Fisher Scientific, Waltham, MA, USA) or Alexa Fluor 568–conjugated goat anti-mouse (1:1000; A11004; Thermo Fisher Scientific, Waltham, MA, USA). Alexa Fluor 488–conjugated streptavidin (S32354; Thermo Fisher Scientific, Waltham, MA, USA) or Alexa Fluor 568-conjugated streptavidin (S11226; Thermo Fisher Scientific, Waltham, MA, USA) was used to detect biotinylated proteins. DNA was detected with Hoechst dye 33342. Coverslips were mounted using 10% (wt/vol) Mowiol 4-88 (Polysciences, Inc., Warrington, PA, USA). Confocal images were obtained using a Nikon A1 confocal microscope (60 ×/1.49 oil APO TIRF Nikon objective) with a charge-coupled device camera (CoolSnap HQ; Photometrics, Tucson, AZ, USA) linked to a workstation running NIS-Elements software (Nikon, Melville, NY, USA). Epifluorescence images were captured using a Nikon Eclipse NiE (20 ×/0.75 Plan Apo Nikon objective) microscope.

### 2.4. Western Blot Analysis

To analyze total cell lysates by immunoblot, 1.2 × 10^6^ cells were lysed in SDS–PAGE sample buffer, boiled for 5 min, and sonicated to shear DNA. Proteins were separated on 4–20% gradient gels (Mini-PROTEAN TGX; Bio-Rad, Hercules, CA, USA) and transferred to nitrocellulose membrane (Bio-Rad, Hercules, CA, US). After blocking with 10% (vol/vol) adult bovine serum and 0.2% Triton X-100 in PBS for 30 min, the membrane was incubated with appropriate primary antibodies: rabbit polyclonal anti-hemagglutinin (1:2000; Ab9110; Abcam, Cambridge, UK) and mouse monoclonal anti-tubulin as a loading control (1:10000; sc-32293; Santa Cruz Biotechnology, Dallas, TX, USA). The primary antibodies were detected using horseradish peroxidase (HRP)–conjugated anti-rabbit (1:40,000; G21234; Thermo Fisher Scientific, Waltham, MA, USA) or anti-mouse (1:40,000; F21453; Thermo Fisher Scientific, Waltham, MA, USA) antibodies. The signals from antibodies were detected using enhanced chemiluminescence via a Bio-Rad ChemiDoc MP System (Bio-Rad, Hercules, CA, USA). Following detection of each antibody, the membrane was quenched with 30% H_2_O_2_ for 30 min. To detect biotinylated proteins, the membrane was incubated with HRP-conjugated streptavidin (1:40,000; ab7403; Abcam, Cambridge, UK) in 0.2% Triton X-100 in PBS for 45 min.

### 2.5. BioID Pulldowns

Large-scale BioID pulldowns were performed as described in [35]. For large-scale TurboID pulldowns, 2 10-cm dishes at 80% confluency were incubated with 50 μM biotin for 18 h and 2 10-cm dishes at 100% confluency were incubated with 50 μM biotin for either 0 or 10 min. From there, pulldowns were performed as described in [35]. Briefly, cells were lysed in 8 M urea 50 mM Tris pH 7.4 containing protease inhibitor (87785: Thermo Fisher Scientific, Waltham, MA, USA) and DTT, incubated with universal nuclease (88700: Thermo Fisher Scientific, Waltham, MA, USA), and sonicated to further shear DNA. Lysates were precleared with Gelatin Sepharose 4B beads (17095601; GE Healthcare, Chicago, IL, USA) for 2 h and then incubated with Streptavidin Sepharose High Performance beads (17511301: GE Healthcare, Chicago, IL, USA) for 4 h. Streptavidin beads were washed four times with 8 M urea 50 mM Tris pH 7.4 wash buffer and resuspended in 50 mM ammonium bicarbonate with 1 mM biotin.

### 2.6. Protein Digestion

Beads were thawed and resuspended with 8 M urea, 50 mM ammonium bicarbonate, and cysteine disulfide bonds were reduced with 10 mM tris(2-carboxyethyl)phosphine (TCEP) at 30 °C for 60 min and cysteines were then alkylated with 30 mM iodoacetamide (IAA) in the dark at room temperature for 30 min. Following alkylation, urea was diluted to 1 M urea, and proteins were subjected to overnight digestion with mass spec grade Trypsin/Lys-C mix (Promega, Madison, WI, USA). Finally, beads were pulled down and the solution with peptides collected into a new tube. The beads were then washed once with 50 mM ammonium bicarbonate to increase peptide recovery. 

Following digestion, samples were acidified with formic acid (FA) and subsequently desalted using AssayMap C18 cartridges (Agilent, Santa Clara, CA, USA) mounted on an Agilent AssayMap BRAVO liquid handling system. Briefly, C18 cartridges were first conditioned with 100% acetonitrile (ACN), followed 0.1% FA. Sample was then loaded onto the conditioned C18 cartridge, washed with 0.1% FA, and eluted with 60% ACN, 0.1% FA. Finally, the organic solvent was removed in a SpeedVac concentrator prior to LC-MS/MS analysis.

### 2.7. Liquid Chromatography (LC) and Mass Spectrometry (MS) Analysis

Dried peptide samples were reconstituted with 2% ACN-0.1% FA and quantified by NanoDrop^TM^ spectrophometer (Thermo Fisher Scientific, Waltham, MA, USA) prior to LC-MS/MS analysis using a Proxeon EASY nanoLC system coupled to a Q-Exactive Plus mass spectrometer (Thermo Fisher Scientific, Waltham, MA, USA). Peptides were separated using an analytical C18 Acclaim PepMap column (75 µm × 250 mm, 2 µm particles; Thermo Fisher Scientific, Waltham, MA, USA) at a flow rate of 300 µL/min using a 118-min gradient: 1% to 6% B in 1 min, 6% to 23% B in 72 min, and 23% to 34% B in 45 min (A = FA, 0.1%; B = 80% ACN: 0.1% FA). The mass spectrometer was operated in positive data-dependent acquisition mode. MS1 spectra were measured with a resolution of 70,000 (AGC target: 1e6; mass range: 350–1700 m/z). Up to 12 MS2 spectra per duty cycle were triggered, fragmented by HCD, and acquired with a resolution of 17,500 (AGC target 5e4, isolation window; 1.2 m/z; normalized collision: 32). Dynamic exclusion was enabled with a duration of 25 s.

### 2.8. Data Analysis

All mass spectra from were analyzed with MaxQuant software version 1.5.5.1. MS/MS spectra were searched against the *Homo sapiens* Uniprot protein sequence database (version January 2018) and GPM cRAP sequences (commonly known protein contaminants). Precursor mass tolerance was set to 20 ppm and 4.5 ppm for the first search where initial mass recalibration was completed and for the main search, respectively. Product ions were searched with a mass tolerance 0.5 Da. The maximum precursor ion charge state used for searching was 7. Carbamidomethylation of cysteines was searched as a fixed modification, while oxidation of methionines and acetylation of protein N-terminal were searched as variable modifications. Enzyme was set to trypsin in a specific mode and a maximum of two missed cleavages was allowed for searching. The target-decoy-based false discovery rate (FDR) filter for spectrum and protein identification was set to 1%. Proteins were classified as candidate interactors if they were identified in all three triplicate samples and abundances were at least 10-fold greater compared to respective controls. The STRING database (www.string-db.org) was utilized for visualizing protein interaction clusters and cellular component GO enrichment analysis. The Retrieve/ID Mapping tool was utilized at www.UniProt.org for subcellular location designations of identified candidate proteins. 

## 3. Results

### 3.1. Comparison of BioID, TurboID, and miniTurbo in Live Cells 

In order to compare BioID, TurboID, and miniTurbo expression and biotinylation in a physiologically relevant cellular setting, tandem HA-tagged (3xHA) versions of each promiscuous ligase were stably expressed in A549 human lung adenocarcinoma cells via retroviral transduction. In an attempt to avoid toxicity issues due to protein overexpression, ligase expression was driven by a retroviral LTR promoter. Cells were assessed via immunofluorescence (IF) and cell lysates by western blot (WB) with or without the addition of 50 µM biotin for 18 h, a typical labeling period for BioID. There was considerable biotinylation in TurboID-only cells without the addition of biotin, as much or more than for BioID after 18 h of biotin supplementation, which is suggestive of a practical lack of inducibility of biotinylation. MiniTurbo-only expressing cells only appeared to promiscuously biotinylate substantially following the addition of biotin for 18 h (Figure 1) and did not appear to biotinylate following 10 min incubation (Appendix A) even in cells expressing similar levels of ligase (Appendix A). However, for the majority of cells, only an extremely low level of miniTurbo protein was detected suggestive of protein instability and/or cell toxicity. Toxicity of TurboID has been previously reported in mammalian cells, worms and flies [34]. Due to the apparent instability of the ligase, miniTurbo was excluded from further study.

Previous reports demonstrated no difference in overall biotinylation levels utilizing TurboID ligase supplemented with either 50 µM or 500 µM biotin [34]. To confirm these reports, A549 cells expressing TurboID ligase were treated with either 50 µM or 500 µM biotin for 10 min, 4 h, or 18 h (Appendix A). Our data support the previous findings as we also observed no substantial difference in biotinylation levels in cells treated with 50 µM or 500 µM biotin at any timepoint. Since the standard biotin concentration for most published BioID studies is 50 µM, and because our current data further supports a 500 µM biotin concentration as unnecessary, we utilized 50 µM biotin for all further studies.

In order to compare the practical application of BioID and TurboID in a cellular setting, we generated several A549 cell lines stably expressing a variety of BioID and TurboID fusion proteins with a tandem HA tag (3xHA). Each ligase was fused to the N-terminus of LaA, the C-terminus of Nup43, and the N-terminus of Nup53. Despite lower levels of expression of TurboID fusion proteins compared to BioID fusion proteins, all cell lines expressing TurboID fusion proteins were promiscuously biotinylating without the addition of excess biotin at levels similar to BioID cells that were induced to biotinylate for 18 h with biotin supplementation (Figure 2).

### 3.2. Application of TurboID to the Nuclear Lamina

To better define the proximity labeling properties of TurboID compared to BioID [6], as comparative proteomic studies of these ligases fused to a specific bait have not been reported, TurboID-LaA and BioID-LaA cells were established and processed in parallel for biotin affinity-purification. A549 stable cell lines expressing each fusion protein were validated via WB and IF (Figure 3). Western blot analysis showed similar levels of biotinylation between BioID-LaA (18 h) and TurboID-LaA without the addition of excess biotin. Furthermore, the addition of excess biotin for 10 min did not significantly increase overall biotinylation. In the absence of exogenous biotin, biotinylation along the nuclear periphery and diffusely in the nucleoplasm of TurboID-LaA cells could be detected by IF with similar localization and intensity to BioID-LaA cells supplemented with biotin for 18 h. Supplementing TurboID-LaA cells with biotin for 10 min slightly increased the biotinylation intensity within the nucleus, whereas an 18 h incubation with excess biotin resulted in apparent saturation of labeling in the nucleus and robust labeling of cytoplasmic proteins. 

To identify which proteins were biotinylated by TurboID and BioID we performed biotin-affinity pulldowns on both sets of cells in triplicate with TurboID-only or BioID-only expressing cells as controls to exclude proteins that non-specifically associate with and/or are biotinylated by the ligases. BioID cells received conventional biotin supplementation for 18 h and TurboID cells received biotin supplementation for 10 min. Captured proteins were identified via mass spectrometry (MS) and analyzed for relevance to the nuclear envelope and nuclear lamina. Protein label-free quantification (LFQ) intensities were summed for each set of triplicates, and only proteins enriched more than 10-fold compared to the respective control and detected in all three triplicates were considered candidate interactors. With these stringent criteria, BioID-LaA identified 58 protein candidates and TurboID-LaA identified 157 protein candidates (Appendix A). The UniProt IDs were utilized to assign subcellular location from the UniProt database. Of the 58 total proteins, 36 (62%) identified in the BioID group were designated as nuclear, compared to 65 out of the 157 proteins (41%) identified in the TurboID group. Of those proteins remaining, 7 were designated as endoplasmic reticulum in the BioID group, compared to 29 proteins in the TurboID group. The remaining proteins (15 for BioID and 63 for TurboID) were given other designations (e.g., cytoplasmic, mitochondrial, cell membrane, etc.) or had no designation by the UniProt database (Appendix A, Figure 4A). STRING-DB analysis revealed several additional clusters in the TurboID-LaA candidates compared to those identified by BioID-LaA, including a wider distribution of cellular component groups (*p* < 0.01), with endoplasmic reticulum groups ranking higher comparatively and cytoplasmic and Golgi groups significantly enriched (*p* < 0.01) (Figure 4B, Appendix A).

### 3.3. Comparing Labeling Radii of BioID and TurboID

The TurboID-LaA results suggested the potential for a larger practical labeling radius than found for BioID. To more directly compare the labeling radii of TurboID to BioID we utilized a similar approach that was used to define the practical labeling radius of BioID and BioID2 [5,13]. Each ligase was fused to the nuclear pore complex (NPC) protein Nup43, a stable constituent of the well-characterized Nup107-160 Y-complex that was previously used as a molecular ruler to estimate the practical labeling radius of BioID and BioID2 [5,13], or Nup53 (also known as Nup35), a stable constituent of the inner ring Nup93 complex which was used to show specificity of biotinylation within the NPC based on its distinct localization in a separate protein complex within the much larger NPC [13]. A549 cells stably expressing BioID- or TurboID- Nup43 or Nup53 were validated via WB and IF (Figure 5) before performing BioID pulldown in triplicate. While WB analysis showed fusion protein expression of both Nup43-TurboID and TurboID-Nup53, protein expression of both ligases was difficult to detect via IF, perhaps due to antibody accessibility and/or low levels of expression (Figure 5). Notably, the IF detection of TurboID was extremely difficult likely due to lower levels of the fusion protein. However, biotinylation activity was robust and clearly labeled the NE for BioID after 18 h of biotin supplementation, and the NE and nucleoplasm for TurboID, both before and after 10 min of biotin supplementation. After 18 h of TurboID labeling the biotinylation was detected throughout the nucleus and into the cytoplasm, and detection of the fusion protein was decreased. Based on these results we supplemented BioID-expressing cells with excess biotin for 18 h and TurboID-expressing cells for 10 min before lysis and biotin-affinity pulldown. Proteins identified via MS were analyzed as described above with all experiments performed in triplicate. As expected, the TurboID radius was demonstrably larger compared to its BioID counterparts, with Nup43-TurboID identifying several more proteins outside the Nup107 complex and throughout the NPC compared to Nup43-BioID (Figure 6A,B, Appendix A). Furthermore, TurboID-Nup53 labeled several more outer components of the NPC compared to BioID-Nup53 (Figure 6C, Appendix A). These findings suggest that TurboID exhibits a larger labeling radius with 10 min of labeling than does BioID in 18 h, albeit a labeling radius that is difficult to measure as is appears to extend beyond the length of the ~35 nm Nup107–160 complex molecular ruler. Furthermore, the IF data suggests that the radius would be expected to dramatically increase with additional time of biotinylation incubation. 

### 3.4. Time Course Comparisons

To determine the practical inducibility of biotinylation by TurboID and to test whether labeling radius expansion correlates with length of biotinylation, the TurboID-Nup53 cells were utilized for BioID pulldown and MS analysis at three timepoints, 0 min, 10 min, and 18 h of biotin supplementation, and were compared to a standard BioID-Nup53 with 18 h of biotin supplementation. After analysis, the MS data revealed extensive overlap in the 0 min sample compared to the TurboID-Nup53 (10 min) sample, revealing that the vast majority of TurboID-Nup53 promiscuous biotinylation at 10 min occurred prior to biotin supplementation (Appendix A). Furthermore, STRING-DB analysis demonstrated an increase in seemingly irrelevant background proteins and a decrease in the overall significance of proteins designated ‘nuclear pore’ and ‘nuclear envelope’ with biotin supplementation and incubation suggesting an expanding radius, which could be detrimental to data analysis and true candidate identification (Figure 7, Appendix A). All four pulldowns ultimately identified the same 22 NE proteins, with slight variation in other NE proteins identified between the BioID-Nup53, TurboID-Nup53 (0 min), and TurboID-Nup53 (10 min) pulldowns.

### 3.5. Transient Expression of TurboID Does Not Substantially Improve Persistent Biotinylation

The observed lack of practical induction of biotinylation in the TurboID cells was considerably different than prior reports which used transient transfection to express TurboID [34]. We hypothesized that perhaps the transient transfection approach prevented a slow accumulation of biotinylated proteins over time that would be expected to occur in cells stably expressing TurboID for many days or weeks. In an attempt to reproduce this rapid induction of biotinylation observed with transient transfection we utilized a mammalian expression vector containing 3xHA-TurboID-NLS (Addgene plasmid #107171) for transient transfection in A549 cells. 24 h after transfection, the cells were either incubated without biotin supplementation or were supplemented with excess biotin for 10 min at 37 °C. WB analysis revealed clear biotinylation without the addition of biotin and a considerable increase in biotinylation following the 10 min biotin supplementation (Figure 8A). Furthermore, IF confirmed biotinylation of proteins within the nucleus as well as throughout the cytoplasm of several cells, both with and without biotin supplementation (Figure 8B). These results support the use of low-expressing, stable cell lines for BioID/TurboID studies to reduce protein mislocalization due to gross and variable overexpression, and although the inducibility of biotinylation is improved compared to stable ligase expression, this suggests a persistent lack of practical inducibility of biotinylation in cells transiently expressing the TurboID ligase.

### 3.6. Intermediate TurboID Variants Exhibited Persistent Biotinylation

During the directed evolution of TurboID and miniTurbo, several intermediate generations were produced with varying degrees of enhanced rate of promiscuous biotinylation [34]. Since TurboID was so highly active at basal biotin levels, we asked whether one of the intermediate generations would enable shorter biotinylation periods without active biotinylation in the absence of biotin supplementation. Each generation contained various mutations (Appendix A) that increased the rate of promiscuous biotinylation [34]. We tested each mutant at various timepoints to assess inducibility and speed of biotinylation (Appendix A). We found that TurboG1 and TurboG2 biotinylation rates were not appreciably enhanced compared to BioID, but TurboG3 did exhibit an increased rate of biotinylation compared to BioID. However, some biotinylation without the addition of biotin was observed, albeit less so than for TurboID. Since biotinylation rates were not significantly enhanced in TurboG1 or TurboG2, and considerable biotinylation in the absence of exogenous biotin was still detected in the TurboG3 cells, we chose not to pursue any further experiments utilizing existing intermediate generations of TurboID.

### 3.7. Removing Biotin from Cell Culture Media Decreases Basal Biotinylation by TurboID

In an attempt to reduce the amount of steady-state biotinylation in cells constitutively expressing TurboID, A549 cells expressing TurboID were cultured in DMEM with 10% dialyzed FBS. Dialysis will remove most of the biotin from the serum, which in our experiments is the sole source of biotin for cells lacking biotin supplementation. Cells were analyzed at 0, 1, 3, 5, and 7 days following culture in dialyzed FBS. As expected, WB analysis showed substantial biotinylation in cells cultured in regular FBS and a decrease of overall biotinylation is evident over the course of 7 days (Appendix A). However, after 7 days in dialyzed FBS the cells still exhibited low levels of biotinylation, presumably due to recycling of the biotin following degradation of previously biotinylation proteins. Since cells are unable to survive without biotin, we did not extend the duration of these depletion studies. We also noticed an increase in the detection of the TurboID by WB, starting at 3 days of biotin depletion, suggesting that the instability of the TurboID results, at least in part, from biotinylation, and possibly from self biotinylation.

### 3.8. Combining Dox-Inducible Expression of TurboID with Dialyzed FBS Allows for Inducible Biotinylation

We sought to ascertain if using an inducible system for protein expression could improve inducible promiscuous biotinylation. Transient transfection typically leads to substantially excessive and highly variable expression levels of exogenous proteins causing subcellular mislocalization and artificial interactions of those fusion proteins, and as such is not recommended for BioID experiments. Therefore, we utilized the retroviral doxycycline (dox)-inducible pRetroX system for stable transduction and inducible expression of 3xHA-TurboID in A549 cells. Following generation of cells with dox-inducible expression of TurboID, biotin was added concurrently for 18 h or for the last 1 h of the 18 h dox incubation of TurboID expression. Cells without dox treatment did not express detectable levels of the TurboID ligase and did not biotinylate following addition of biotin for 1 h, but did produce detectable biotinylation with 18 h biotin incubation, indicating some low-level TurboID expression without the addition of dox (Appendix A). When TurboID expression was induced for 18 h (+dox), cells were once again biotinylating without biotin supplementation, and that biotinylation increased with biotin supplementation. Since the standard dox incubation for inducible expression tends to be 24–48 h and promiscuous biotinylation is evident after 18 h with dox regardless of biotin supplementation, transient induction of TurboID expression does not practically improve the inducibility of TurboID biotinylation under conventional culture conditions.

The dox-inducible system allows for the control of TurboID expression and the dialyzed serum enables tight control over the introduction of biotin. As such, we asked whether combining these two approaches would allow for practical inducibility of promiscuous biotinylation. A549 cells stably expressing dox-inducible TurboID were cultured in DMEM 10% regular FBS until dox-induction. Culture media was removed, washed once with PBS, and cells were replenished with DMEM containing 10% dialyzed FBS and dox and incubated for 18 h. Cells were then supplemented with biotin for 10 min, 1 h, or 4 h and analyzed via WB (Figure 9). TurboID expression was detected only after adding doxycycline and biotinylation was evident after 1 h and, even more so, 4 h following biotin supplementation. Thus, a combination of inducible expression of TurboID in biotin-depleted conditions followed by biotin supplementation appeared to enable practical inducibility of biotinylation by TurboID within an hour.

### 3.9. TurboID in the Endoplasmic Reticulum Lumen

Our previous unpublished studies have revealed that the endoplasmic reticulum (ER) lumen is considerably less conducive to biotinylation via BioID as overall biotinylation of proteins in this compartment is decreased as observed by IF and IB and the number of candidates identified by MS is notably reduced. The specific mechanism(s) limiting BioID biotinylation in the ER remains unclear, but could be due to an ER-resident biotinidase that is de-biotinylating proteins, limited level of biotin as there are no recognized mechanisms to actively transport biotin into the ER lumen, BioID misfolding due to ER resident chaperones and/or redox state, and/or an increase in nucleophile concentration driving elevated bio-AMP hydrolysis. The robust biotinylation demonstrated by TurboID in each of the previous experiments led us to question whether TurboID would be a good alternative to BioID for identifying PPIs in the ER lumen. BioID or TurboID was fused to the lumenal C-terminus of the inner nuclear membrane (INM) protein Sun2 in order to determine biotinylation capabilities within the ER/INM lumen. The fusion proteins were then stably expressed in A549 cells and analyzed via IF and WB with and without biotin addition for different incubation durations. Interestingly, Sun2-TurboID fusion protein was able to be expressed at considerably higher levels compared to the nuclear and cytoplasmic TurboID-only (Figure 10A), and, while still not optimally inducible for biotinylation, TurboID was able to robustly biotinylate proteins within the ER lumen in contrast to BioID (Figure 10B).

## 4. Discussion

Our studies in live cells stably expressing TurboID and miniTurboID further support prior observation of potential toxicity and/or protein instability issues involving TurboID and miniTurbo that were reported in *C. elegans* and *D. melanogaster* [34]. Cells expressing miniTurbo, and to a lesser extent TurboID, exhibited reduced levels of the ligase compared to BioID, possibly due to selection against cells with elevated expression and/or rapid degradation of these ligases. These toxicity and/or instability issues led us to exclude miniTurbo from subsequent studies as conventional PPI experiments in unhealthy cells could confound results. The cellular toxicity of these TurboID and especially miniTurboID proteins could come from persistent biotinylation of various proteins leading to their dysfunction. Another more likely source of toxicity is that these ligases could be competing with the endogenous biotin ligase for access to free biotin necessary to biotinylate key metabolic enzymes, a finding supported by the need for biotin supplementation to enable survival of a fly model stably expressing TurboID [34]. Together, this new data suggests investigators should be cautious about using miniTurbo for proximity-labeling studies.

Our initial data comparing BioID- and TurboID-fused proteins revealed persistent steady-state biotinylation by TurboID under normal cell culture conditions that confounds its use for comparative studies using short labeling periods. In all of our experiments, TurboID persistently biotinylated proximal proteins at levels similar to a conventional induced BioID experiment. These findings emphasize the importance of validating the inducibility of biotinylation in TurboID-expressing cells prior to proximity-labeling experiments to ensure that there is nominal pre-existing biotinylation, and, when relevant, would support the use of no-biotin controls for proteomic experiments to ensure that detected proteins are unique to the exogenous biotin-induced labeling period.

The total number of proteins identified by TurboID-LaA compared to BioID-LaA likely reflects an increased practical labeling radius and could prove problematic when attempting to map candidate PPI networks. We use the term ‘practical’ to reflect the distribution of the candidates identified by MS following a BioID or TurboID pulldown. The labeling radius may actually be increased for TurboID due to the saturation of available biotinylation sites in close proximity to the ligase that could promote labeling of sites further away. Alternatively, this may also reflect that the ligase is always active, albeit at lower levels in a non-biotin supplemented condition, and subpopulations of the bait proteins are in fact associated with these candidates throughout various stages of the cell cycle, perhaps especially during mitosis when there is no nuclear envelope and the nuclear lamina is disassembled. Another possible reason for some of the discrepancy in some of the candidates between BioID-LaA and TurboID-LaA may reflect a genuine difference in association of these proteins due to the cellular toxicity observed with TurboID constructs. Regardless of the reasons, the increase in proteins designated endoplasmic reticulum, cytoplasm, mitochondrion, and all other subcellular locations in the TurboID-identified candidates compared to the BioID-identified candidates forecasts a daunting task of further deciphering which proteins are relevant interactors versus irrelevant background proteins. What is clear however, is that the more abundant proteins detected by TurboID-LaA are most similar to those for BioID-LaA and are more likely to reflect proximate protein associations. The less abundant proteins are the ones that are more disparate.

Our application of TurboID to Nups in the NPC also supported these inducibility of biotinylation and labeling radius issues. Proteomic analysis of TurboID-Nup53 cells at different timepoints following supplementation with biotin and isolation of biotinylated proteins demonstrates that TurboID does indeed label known proximate proteins without the addition of biotin and that further biotin supplementation only further increases the practical labeling radius of TurboID. For this reason, cells constitutively expressing TurboID will be uninducible for biotinylation in the conventional sense—that is they will continuously biotinylate proximate proteins under traditional culture conditions, which may preclude use of this faster ligase from certain experiments requiring short labeling periods, one of the primary motivations for the creation of TurboID.

In order to develop a new protocol that enabled reliable PPI-detection without the data-contamination that would occur as a result of perpetual biotinylation, we assessed several intermediate TurboID variants, tested transient transfection, utilized dox-inducible expression of TurboID, and cultured cells in dialyzed FBS that lacks biotin. We found that of these methods tested, combining inducible expression of TurboID with the use of dialyzed FBS allowed for inducible biotinylation by TurboID. Investigators who seek to compare a single cell line under multiple conditions should consider utilizing this approach to allow for detection of subtle changes or transient interactions that may otherwise be undetectable in a cell with pre-existing biotinylation of proximate proteins. Due to the increasing practical labeling radius that occurs with the length of biotin supplementation, it may be necessary for investigators to perform preliminary time-course experiments to determine the optimal length of biotin incubation as our studies show varying degrees of biotinylation for each bait tested.

Overall, these studies support TurboID as a rapid, promiscuous biotin ligase that can utilize basal levels of biotin in cell culture media for producing reactive bioAMP molecules. TurboID likely shares a similar mechanism of action with BioID, where both ligases create and release these reactive bioAMP molecules, but TurboID clearly does so at a much faster rate likely due, at least in part, to an enhanced affinity to biotin [9,10]. The expanding practical labeling radius evident with increasing biotin concentration and incubation duration may therefore impair the use of TurboID for some common BioID applications, but does not prohibit the use of TurboID under all circumstances. If the investigator is solely interested in elucidating the PPIs of a single bait under a single condition, it may be possible to utilize TurboID culture under normal media conditions and lyse the cells for pull down without supplementing with biotin. Additionally, if a BioID-bait fusion protein expression is extremely low due to toxicity, degradation, or suppression, it may be beneficial to try TurboID as the robust levels of biotinylation should be able to overcome low levels of expression. Furthermore, our studies support TurboID as the clearly superior ligase in the ER lumen with the ability to robustly biotinylate proteins in an environment where BioID is only marginally effective. Perhaps TurboID will be useful for other environments that may affect conventional BioID function such as organelles with low pH or extracellularly. Subsequent studies exploring these possibilities will be necessary to determine just how effectively TurboID will operate in these environments. It is clear that there is still the need for a proximity labeling method for living cells that has a well-defined practical labeling radius and that can be rapidly induced to enable studies of brief temporally distinct differences in protein associations. Whether this comes in the form of further modifications of labeling protocols with the existing ligases, modification to existing ligases, and/or entirely different approaches to proximity protein labeling remains to be seen.

## Figures and Tables

**Figure 1 cells-09-01070-f001:**
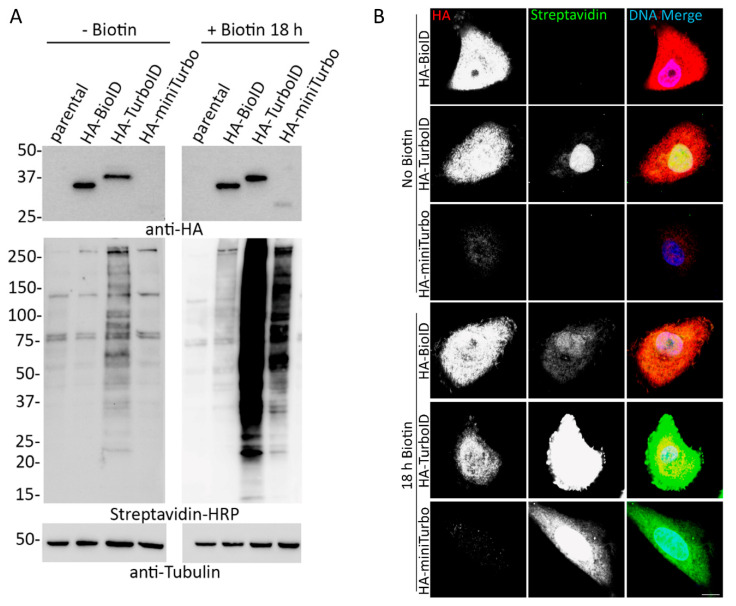
Addition of exogenous biotin increases biotinylation in HA-tagged BioID, TurboID, and miniTurbo expressing cells. (**A**) A549 cell lysates stably expressing HA-tagged BioID, TurboID, or miniTurbo were analyzed via WB in the absence and presence of exogenous biotin (18 h). Anti-HA was used to probe for expression of the ligase and streptavidin HRP was used to probe for biotinylation. Anti-tubulin was used as protein loading control. (**B**) Confocal images of A549 cells stably expressing HA-tagged BioID, TurboID, or miniTurbo with and without the addition of excess biotin for 18 h. Anti-HA was used to visualize ligase localization (red) and fluorescently conjugated streptavidin was used to visualize biotinylation (green). Hoechst dye was used to detect the DNA in the nucleus (blue in merge). Scale bar 10 µm.

**Figure 2 cells-09-01070-f002:**
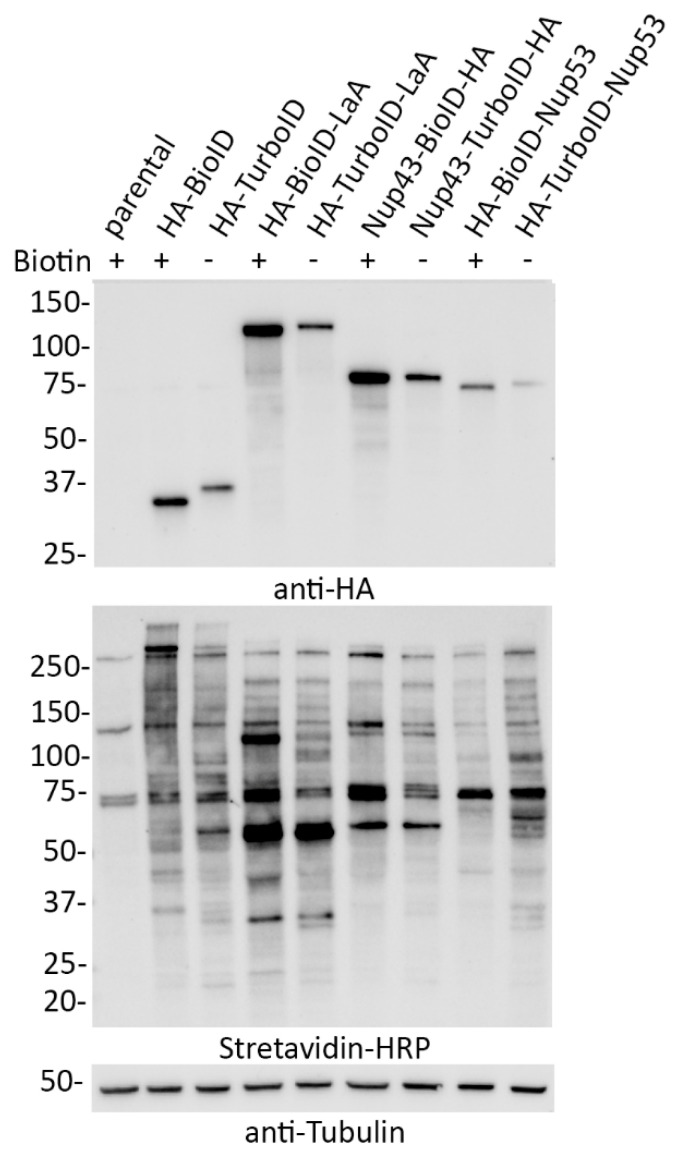
TurboID-fusion proteins biotinylate without the addition of biotin at similar levels to BioID- fusion proteins with biotin supplementation. Whole-cell lysates of A549 cells stably expressing BioID- or TurboID-tagged LaA, Nup43, or Nup53 fusion proteins were analyzed via western blot in the presence (+) of 50 μM biotin supplementation for 18 h or absence (−) of biotin supplementation. Anti-HA was used to detect fusion protein levels and streptavidin-HRP was used to probe for biotinylation. Anti-tubulin was used as a protein loading control.

**Figure 3 cells-09-01070-f003:**
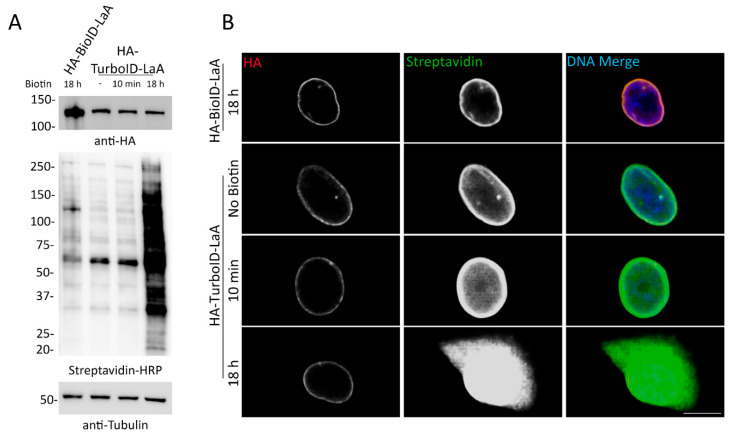
A549 cells stably expressing TurboID-LaA biotinylate without the addition of exogenous biotin. (**A**) Western blot analysis of HA-tagged BioID-LaA and TurboID-LaA fusion protein expression (anti-HA) and overall biotinylation (Streptavidin-HRP) without or after biotin supplementation for 10 min or 18 h. Anti-tubulin was used as a protein loading control. (**B**) Confocal images of A549 cells stably expressing HA-tagged BioID-LaA and TurboID-LaA. Anti-HA was used to visualize fusion protein localization (red). Fluorescently conjugated streptavidin was used to detect biotinylation following no biotin supplementation or addition of biotin for 10 min or 18 h. Hoechst dye was used to detect the DNA in the nucleus (blue in merge). Scale bar 10 µm.

**Figure 4 cells-09-01070-f004:**
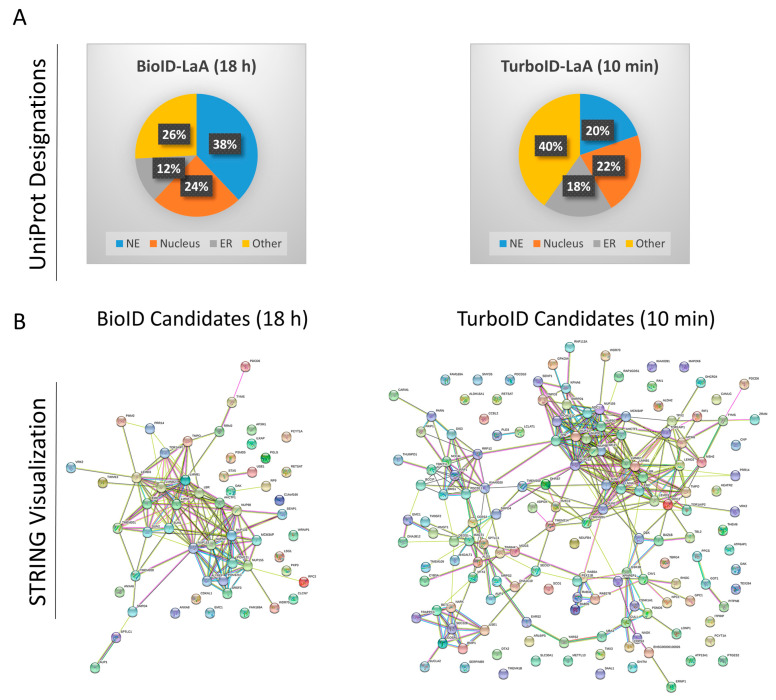
Post-pulldown proteomic analysis of LaA PPI candidates. (**A**) Following MS data analysis, BioID- and TurboID-LaA candidates were submitted to UniProt using the “Retrieve/ID Mapping” tool and subsequently grouped by subcellular domain designations. (**B**) Candidates were also submitted for STRING analysis visualization.

**Figure 5 cells-09-01070-f005:**
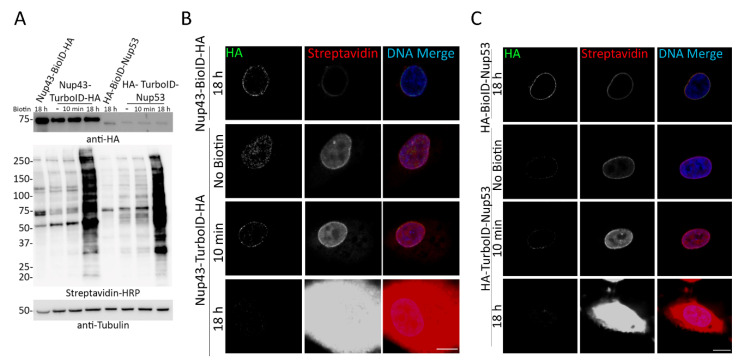
Nup43-TurboID-HA and HA-TurboID-Nup53 biotinylate without the addition of biotin. (**A**) Western blot analysis of A549 cell lysates expressing BioID- or TurboID-fused Nup43 or Nup53 in the absence or presence (10 min or 18 h) of biotin supplementation. Anti HA was used to detect fusion protein expression and streptavidin HRP was used to detect biotinylation. Anti-tubulin was used as a protein loading control. (**B**,**C**) Confocal visualization of BioID- or TurboID- Nup43 and Nup53. Fluorescently conjugated streptavidin was used to visualize biotinylation (green) following the addition (10 min or 18 h) or absence of 50 µM biotin. Anti-HA was used to visualize fusion protein expression and localization (red). Hoechst dye was used to detect the DNA in the nucleus (blue in merge). Scale bar 10 µm.

**Figure 6 cells-09-01070-f006:**
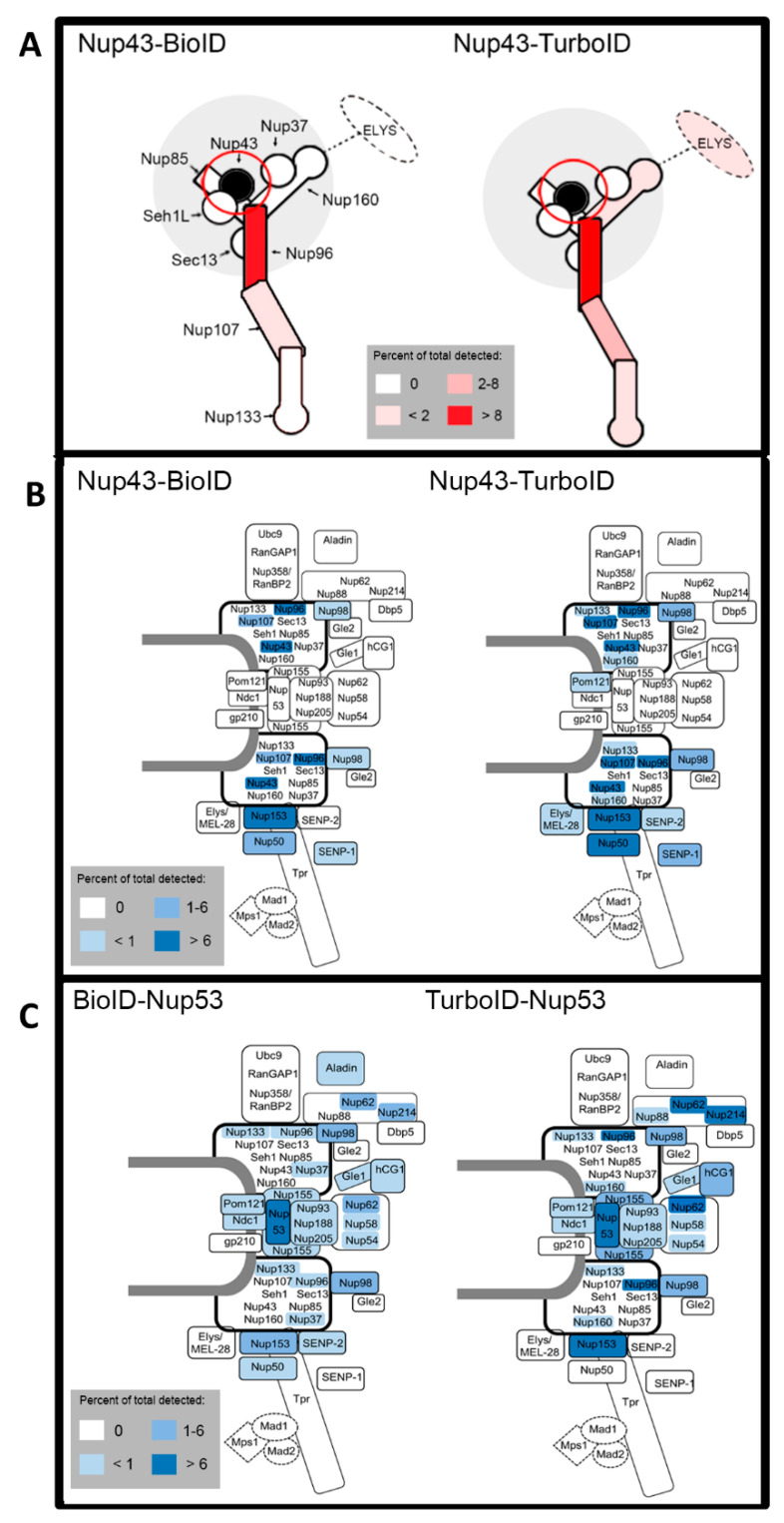
Schematics of practical labeling radii comparisons for Nup43 and Nup53 fusion proteins. (**A**) Schematic of labeling radii of Nup43-BioID and Nup43-TurboID within the Nup107-160 complex. Intensity of red labeling correlates with normalized percent total detected for each identified protein. (**B**) Schematic of labeling radii of Nup43-BioID and Nup43-TurboID within the NPC. Intensity of blue labeling correlates with normalized percent total detected for each identified protein. (**C**) Similar schematic as (**B**) for BioID-Nup53 and TurboID-Nup53.

**Figure 7 cells-09-01070-f007:**
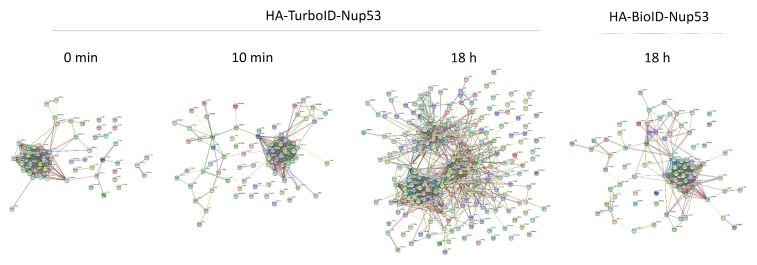
Major candidates for TurboID-Nup53 are biotinylated prior to biotin supplementation. Protein IDs for each ligase and timepoint were submitted to the STRING database for cluster visualization.

**Figure 8 cells-09-01070-f008:**
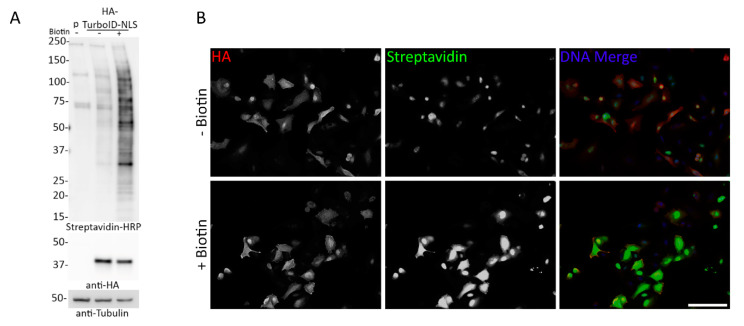
Transient transfection of TurboID in A549 cells does not improve inducibility of biotinylation. (**A**) A549 cells transiently transfected with HA-TurboID-NLS were analyzed for expression (anti-HA) and biotinylation (Streptavidin-HRP) in the presence and absence of biotin (10 min) via WB. 24 h after transient transfection, cells were exposed to biotin for 10 min. Anti-tubulin was used as a protein loading control. (**B**) Epifluorescence images of A549 cells transiently transfected with HA-TurboID-NLS. Anti-HA was used to detect protein localization and fluorescently conjugated streptavidin was used to visualize biotinylation. Hoechst dye was used to detect the DNA in the nucleus (blue in merge). Scale bar 10 µm.

**Figure 9 cells-09-01070-f009:**
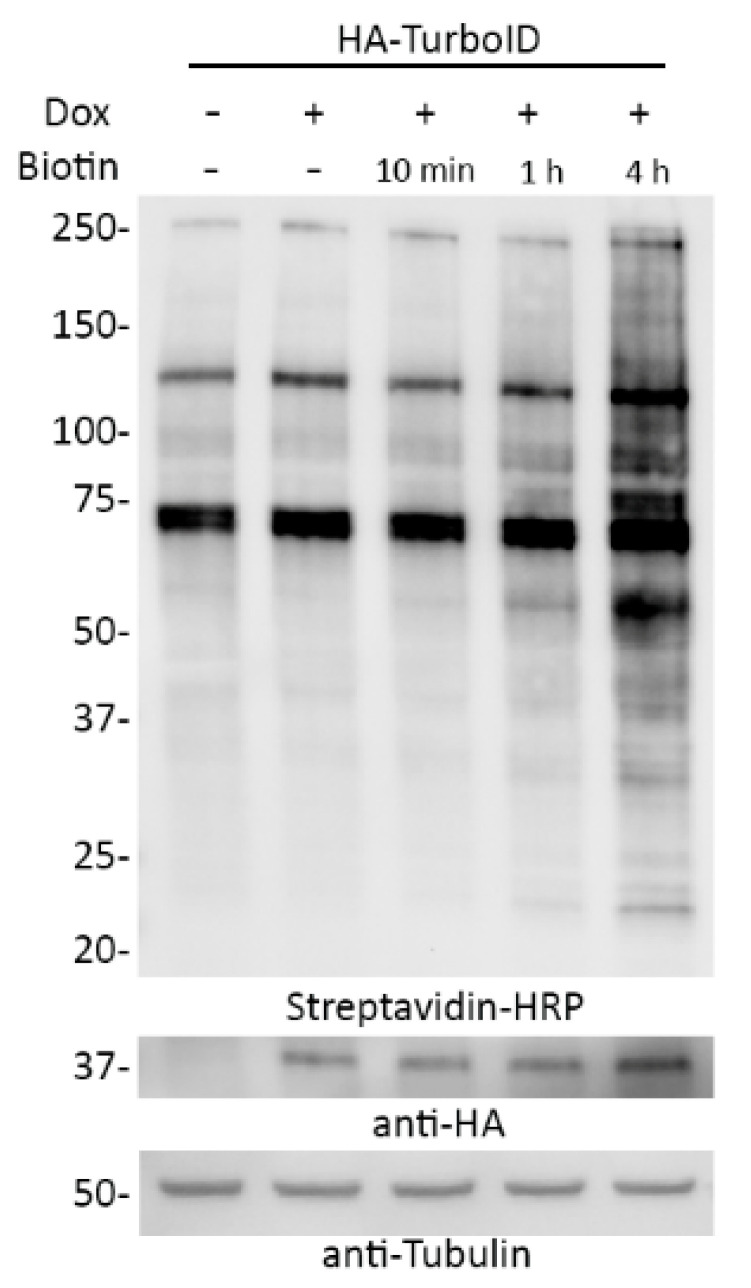
Inducible biotinylation can be achieved by combining dox-inducible TurboID expression with growth in biotin-depleted media. A549 cells stably expressing dox inducible HA-tagged TurboID were incubated with doxycycline and DMEM with 10% dialyzed serum for 18 h prior to the addition of biotin. The cells were then incubated with biotin for either 10 min, 1 h, or 4 h. Streptavidin-HRP was used to visualize biotinylation levels. Anti-HA was used to observe fusion protein levels. Anti-tubulin was used as a protein loading control.

**Figure 10 cells-09-01070-f010:**
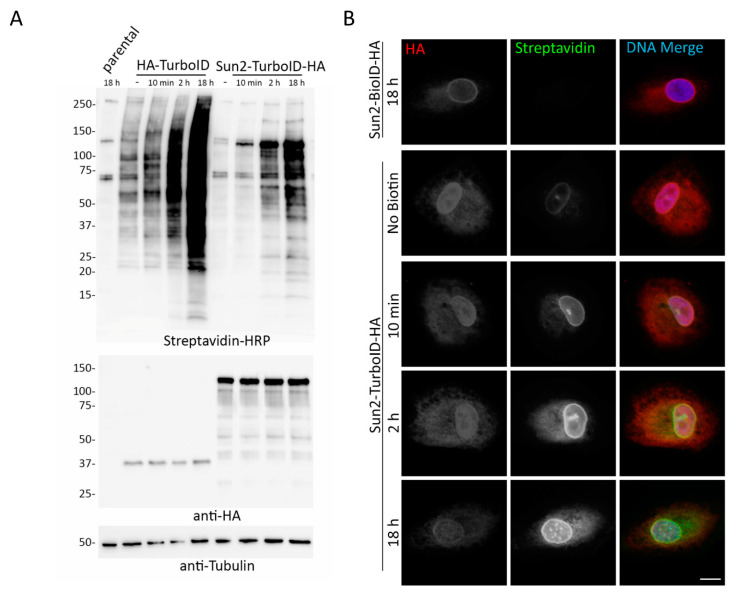
TurboID is relatively stable within the ER lumen and biotinylates more efficiently than BioID. (**A**) Western blot analysis of A549 cells stably expressing HA-TurboID or HA-TurboID-Sun2 in the absence or presence (10 min, 2 h, or 18 h). A549 parental cells incubated with biotin for 18 h was used as a control for basal biotinylation levels. (**B**) Sun2-BioID-HA showed no biotinylation following 18 h of exogenous biotinylation, while Sun2-TurboID-HA showed biotinylation in the absence and presence (10 min, 2 h, 18 h) of incubation with excess biotin. Anti-HA (red) was used to visualize fusion protein localization and fluorescently conjugated streptavidin (green) was used to visualize biotinylation. Hoechst dye was used to detect the DNA in the nucleus (blue in merge). Scale bar 10 µm.

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
