# Peer review of "Comparative Application of BioID and TurboID for Protein-Proximity Biotinylation"

_cells, 2020, doi:10.3390/cells9051070_

Round 1
Reviewer 1 Report
The manuscript from May et al aims at assessing the suitability of TurboID use for BioID experiments. The authors compare the effectiveness of TurboID, and to a lesser extend miniTurbo, as a replacement of an earlier BioID variant developed by Roux et al. In comparison to TurboID and miniTurbo, earlier versions indeed require long labelling periods, which precludes their effective use in time constrained studies.
May and colleagues first compare the expression and biotinylation capabilities of TurboID, miniTurbo and BioID. Secondly, they define the experimental conditions in which TurboID can be used as a substitute for BioID, while retaining biotinylation characteristics suitable for experiments; both in term of undesirable background and expected specific biotinylation.
Overall, the manuscript is well written and with a few alterations can therefore be published.
1) The comparison in of miniTurbo, TurboID and BioID is important but slightly cut short with the exclusion of miniTurbo from further experiments. The authors submit that miniTurbo is toxic or triggers cell toxicity hence do not investigate miniTurbo further, but the underlying data is not shown in the manuscript (data not shown). How does the toxicity of miniTurbo compares to that of BioID, how does it compare to TurboID itself, and most interestingly since the authors focus thereafter on TurboID, how does TurboID potential toxicity compares to the standard BioID in their experimental setup?
Extensive testing to determine whether these results could be construct or cell line specific (fig 1 only expressed unfused ligases in a single cell line) could be beyond the scope of this article, but a simple growth curve using the very cells used in fig1 should be able to make a stronger case. The authors do suggest that miniTurbo show expression defects, but according to the data this is also clearly apparent for all TurboID constructs used in fig2, fig3 and in fig5, and indeed mentioned in the main text in several places. Such a comparison would also help with discarding miniTurbo per se, as its response to biotin supplementation and background biotinylation levels appear much less problematic than that of TurboID.
2) The interest of the STRING visualization in fig4B is slightly offset by the fact that protein names are no readable, a full resolution supplement figure could address this. Since the point of the panel seems to be showing the increase in clusters with TurboID, another possibility is to make sure that shared proteins clusters appear at the same location between BioID and TurboID panels. As currently depicted, there is little possibility to visually compare the two meaningfully, as there is no way of knowing where are the additional or shared clusters, and mere numbers would achieve the very same result.
3) In figure 6B-C some font colours vary, potentially according to the enrichment of the stated protein. Confusingly, some variations pertain to subcomplexes already colour coded but still including variable fonts, with coloured outlines in some cases, without clear mention of which supersedes the other (if any).
4) “Inducibility” is here used interchangeably to refer to changes in expression levels, or response to biotin supplementation (actually occurring under a constant amount of available biotin ligase activity). “Response to biotin supplementation”, or equivalent could be a slightly less confusing alternative. Similarly line 412, instead of “A549 cells stably expressing dox-inducible TurboID”, a statement such as “A549 stably transduced with” should be a more accurate.
Author Response
Response to Reviewer #1
The manuscript from May et al aims at assessing the suitability of TurboID use for BioID experiments. The authors compare the effectiveness of TurboID, and to a lesser extend miniTurbo, as a replacement of an earlier BioID variant developed by Roux et al. In comparison to TurboID and miniTurbo, earlier versions indeed require long labelling periods, which precludes their effective use in time constrained studies.
May and colleagues first compare the expression and biotinylation capabilities of TurboID, miniTurbo and BioID. Secondly, they define the experimental conditions in which TurboID can be used as a substitute for BioID, while retaining biotinylation characteristics suitable for experiments; both in term of undesirable background and expected specific biotinylation.
Overall, the manuscript is well written and with a few alterations can therefore be published.
1) The comparison in of miniTurbo, TurboID and BioID is important but slightly cut short with the exclusion of miniTurbo from further experiments. The authors submit that miniTurbo is toxic or triggers cell toxicity hence do not investigate miniTurbo further, but the underlying data is not shown in the manuscript (data not shown). How does the toxicity of miniTurbo compares to that of BioID, how does it compare to TurboID itself, and most interestingly since the authors focus thereafter on TurboID, how does TurboID potential toxicity compares to the standard BioID in their experimental setup?
Extensive testing to determine whether these results could be construct or cell line specific (fig 1 only expressed unfused ligases in a single cell line) could be beyond the scope of this article, but a simple growth curve using the very cells used in fig1 should be able to make a stronger case. The authors do suggest that miniTurbo show expression defects, but according to the data this is also clearly apparent for all TurboID constructs used in fig2, fig3 and in fig5, and indeed mentioned in the main text in several places. Such a comparison would also help with discarding miniTurbo per se, as its response to biotin supplementation and background biotinylation levels appear much less problematic than that of TurboID.
Our findings on the cellular toxicity of TurboID and miniTurbo were entirely observational during our generation and utilization of the cells with proliferation and passaging rates being obviously impacted. However, we did not specifically measure those impacts and would practically be unable to do so with the rapid turnaround for the revisions. To obtain the data needed to demonstrate cell toxicity of TurboID and miniTurbo that has already been reported, we would need to perform additional experiments that would considerably delay dissemination of our findings, therefore we have slightly modified the text of our manuscript to focus on protein instability and previous reports of toxicity. Regarding our decision to abandon miniTurbo early in the process, this was based on the combination of not only the apparent toxicity to the cells, but also the profoundly limited expression of the protein in cells when stably expressed, to an extent far greater than observed for TurboID. We did adjust the text in this regard to more clearly support that our decision and that our findings are similar to those reported in the original TurboID and miniTurboID studies.
2) The interest of the STRING visualization in fig4B is slightly offset by the fact that protein names are no readable, a full resolution supplement figure could address this. Since the point of the panel seems to be showing the increase in clusters with TurboID, another possibility is to make sure that shared proteins clusters appear at the same location between BioID and TurboID panels. As currently depicted, there is little possibility to visually compare the two meaningfully, as there is no way of knowing where are the additional or shared clusters, and mere numbers would achieve the very same result.
We agree that it would be highly beneficial for readers to be able to zoom in on the STRING figure and read individual protein names. The figure we uploaded is indeed of sufficient resolution to allow this, and would need to be made available online for readers to download as it will be unreadable in the manuscript itself. We believe the current STRING visualization adequately shows an increase in the number of proteins labeled over time as each TurboID network grows with biotinylation duration. For those who wish to look more closely at individual “clusters,” we included Table S6 which lists more detail.
3) In figure 6B-C some font colours vary, potentially according to the enrichment of the stated protein. Confusingly, some variations pertain to subcomplexes already colour coded but still including variable fonts, with coloured outlines in some cases, without clear mention of which supersedes the other (if any).
We agree that this combination of font color and shaded boxes was confusing. It was meant to highlight proteins within complexes that were detected, so the entire complex was shaded and the font color indicated the detected proteins. To make this more clear and straightforward we simply shaded the region around each protein within a complex and left the font colors unchanged.
4) “Inducibility” is here used interchangeably to refer to changes in expression levels, or response to biotin supplementation (actually occurring under a constant amount of available biotin ligase activity). “Response to biotin supplementation”, or equivalent could be a slightly less confusing alternative. Similarly line 412, instead of “A549 cells stably expressing dox-inducible TurboID”, a statement such as “A549 stably transduced with” should be a more accurate.
We agree that this it is confusing to sometimes refer to inducible/inducibility in the terms of biotinylation and other times protein expression. We made sure to clearly indicate our intent throughout the manuscript.
Reviewer 2 Report
In this study, May et al compared two methods for proximity-dependent labeling, BioID and TurboID. A series of well controlled experiments were performed using HA-tagged constructs as well as three fusion constructs, i.e. LaA, Nup43, and Nup53. These experiments led to the conclusions that TurboID can work even without addition of biotin, probably due to its significantly enhanced activities and therefore work when endogenous biotin level is low. Additionally, TurboID may have extended labeling radii and capture more neighborhood proteins. However, it remains to be determined whether or not this is advantageous or it may introduce more background signals. Moreover, the authors showed that TuroboID works well in the ER lumen.
Overall, this is a very nice technical paper which directly compared two existing technologies. There are some minor weaknesses that need to be at least discussed. First, the miniTurbo seems to be a good tool, since it appears to have low expression, low background, but greatly enhanced signals following biotin incubation. It is not clear why the authors decided not to further compare it with their BioID method. Second, the advantage of short labeling (10 mins versus 18 hours) is to capture transient interactions. The authors should test or at least discuss the possibility using TurboID for studying transient protein-protein interactions, such as following stimulation with growth factors or others.
Author Response
Response to Reviewer #2
In this study, May et al compared two methods for proximity-dependent labeling, BioID and TurboID. A series of well controlled experiments were performed using HA-tagged constructs as well as three fusion constructs, i.e. LaA, Nup43, and Nup53. These experiments led to the conclusions that TurboID can work even without addition of biotin, probably due to its significantly enhanced activities and therefore work when endogenous biotin level is low. Additionally, TurboID may have extended labeling radii and capture more neighborhood proteins. However, it remains to be determined whether or not this is advantageous or it may introduce more background signals. Moreover, the authors showed that TuroboID works well in the ER lumen.
Overall, this is a very nice technical paper which directly compared two existing technologies. There are some minor weaknesses that need to be at least discussed. First, the miniTurbo seems to be a good tool, since it appears to have low expression, low background, but greatly enhanced signals following biotin incubation. It is not clear why the authors decided not to further compare it with their BioID method.
We have tried to more clearly articulate our rationale for this decision as being a combination of the negative impact of miniTurboID on the growth of the cells, in combination with the apparent instability of the protein, both of which could lead to unhealthy/aberrant cell behavior. We discuss this in the first paragraph of the discussion.
Second, the advantage of short labeling (10 mins versus 18 hours) is to capture transient interactions. The authors should test or at least discuss the possibility using TurboID for studying transient protein-protein interactions, such as following stimulation with growth factors or others.
We do not believe that TurboID excels over BioID in capturing transient interactions per se, as they both are clearly capable of doing that. The longer acting BioID would simply accumulate a history of those transient interactions over a longer period of time. We do agree that a more active ligase like TurboID would be capable of capturing interactions during a more transient labeling period (e.g. during an induced biological process like DNA-damage). However, for this to be possible there would have to be no basal biotinylation by TurboID, something that was not possible in our experiments without inducible expression of the ligase combined with depletion of basal biotin from the cells prior to that induced expression. Hopefully, other investigators can use this information to more appropriately design their experiments with TurboID. This was discussed in the 5th paragraph of the discussion.
Reviewer 3 Report
In this manuscript the authors provide a valuable comparison between BioID and TurboID. These methods are used extensively, therefore a proper comparison is very handy and important to the field. I believe the experiments and analysis were done in a very elegant manner. In the future, tt would be of a major importance to use the experimental system described here (lamin A) and correlate it with Cryo-ET in order to understand how close the biolinylated protein to the lamina and weather the BirA at the n-terminus would impose structural changes in lamin filaments or organization of the lamina.
Author Response
Response to Reviewer #3
In this manuscript the authors provide a valuable comparison between BioID and TurboID. These methods are used extensively, therefore a proper comparison is very handy and important to the field. I believe the experiments and analysis were done in a very elegant manner. In the future, tt would be of a major importance to use the experimental system described here (lamin A) and correlate it with Cryo-ET in order to understand how close the biolinylated protein to the lamina and weather the BirA at the n-terminus would impose structural changes in lamin filaments or organization of the lamina.
Thank you for your suggestion for future studies. Since there are both nuclear lamina and nucleoplasmic population of lamins, perhaps the more stable and discretely localized NPC proteins would be a better choice for the ultrastructural analysis. And we agree that the addition of a GFP-sized protein to the lamins, or any other protein is always a cause for concern as to the function of the protein.